# Label-Free Anomaly Detection Using Distributed Optical Fiber Acoustic Sensing

**DOI:** 10.3390/s23084094

**Published:** 2023-04-19

**Authors:** Yuyuan Xie, Maoning Wang, Yuzhong Zhong, Lin Deng, Jianwei Zhang

**Affiliations:** 1Sichuan University National Key Laboratory of Fundamental Science on Synthetic Vision, Sichuan University, Chengdu 610064, China; 2College of Electrical Engineering, Sichuan University, Chengdu 610065, China

**Keywords:** deep learning, distributed optical fiber acoustic sensing, unsupervised learning

## Abstract

Deep learning anomaly detection is important in distributed optical fiber acoustic sensing (DAS). However, anomaly detection is more challenging than traditional learning tasks, due to the scarcity of true-positive data and the vast imbalance and irregularity within datasets. Furthermore, it is impossible to catalog all types of anomalies, therefore, the direct application of supervised learning is deficient. To overcome these problems, an unsupervised deep learning method that only learns the normal data features from ordinary events is proposed. First, a convolutional autoencoder is used to extract DAS signal features. A clustering algorithm then locates the feature center of the normal data, and the distance to the new signal is used to determine whether it is an anomaly. The efficacy of the proposed method was evaluated in a real high-speed rail intrusion scenario, and considered all behaviors that may threaten the normal operation of high-speed trains as abnormal. The results show that the threat detection rate of this method reaches 91.5%, which is 5.9% higher than that of the state-of-the-art supervised network and, at 7.2%, the false alarm rate is 0.8% lower than the supervised network. Moreover, using a shallow autoencoder reduces the parameters to 1.34 K, which is significantly lower than the 79.55 K of the state-of-the-art supervised network.

## 1. Introduction

Anomaly detection refers to the process of detecting data events that significantly deviate from the norm. Owing to the increasing demand for such applications in domains such as risk management, security, financial surveillance, health, and medicine, a variety of machine-learning anomaly detection methods are being developed [1]. Video data are a common resource for anomaly detection [2,3]. However, three key challenges hamper the deployment of machine learning video-event anomaly solutions. First, deployment is difficult and incurs high costs. Furthermore, systems can only be installed in specific areas and abnormal events can occur anywhere. Hence, blind and dead spots are problematic. Second, troubleshooting and maintenance are difficult and require vast human and material resources. Last, events can be easily disturbed by external factors, such as haze, weather, luminance, and reflections. Distributed optical fiber acoustic sensing (DAS) can solve these problems because it has many advantages, such as ultra-long detection distances, easy deployment, and resistance to harsh environments [4]. Therefore, it is highly suitable for anomaly detection [5].

DAS utilizes Rayleigh backscattering to measure sound or vibrations along an optical fiber over ultra-long distances [6]. Presently, the application of DAS is being studied in several fields, such as pipeline monitoring [7,8,9,10] that uses a buried optical fiber cable next to the pipeline. Perimeter security [11,12,13] is similarly monitored using a DAS system at the boundary of important facilities to detect intrusions. Earthquake detection [14,15,16,17] and other vibrational events are also monitored in this manner.

Numerous studies have focused on improving the performance of DAS, e.g., by applying new transmitted light schemes and scattering phase demodulation. Sensing range performance parameters, e.g., spatial and sensing resolutions, have significantly improved over the years [18]. To improve performance further, DAS methods are being combined with deep learning techniques for the intelligent and real-time identification of vibrations along the optical fiber [19,20,21,22,23].

However, the integration of deep learning inherits the same challenges faced by other machine learning anomaly detection models [1,24]. For example, the extreme imbalance between normal and abnormal samples in sensory datasets reduces the generalizability of the model. Many studies use artificially simulated abnormal data; however, it is impossible to simulate all DAS abnormalities because there is an abundance of noise in real environments. These challenges make it difficult to apply traditional supervised learning methods directly.

To address these challenges, this paper proposes an unsupervised method that only learns normal data features. Because the DAS systems collect multi-channel time series signals, the proposed unsupervised method considers the space and time dimensions. Thus, over a given period the original DAS signal is divided into windows consisting of several nearby channels and becomes the model’s input. Then, an autoencoder is used to extract normal data features, and a clustering algorithm is used to establish feature centers. During testing, if a feature in the window is sufficiently distant from the center, it is judged to be abnormal. Notably, the autoencoder’s shallow convolution structure is time efficient. To evaluate the efficacy of the proposed method, experiments were performed on the data from a real-world high-speed rail intrusion scenario. Intrusion behaviors, such as wall climbing and wall breaking, are regarded as abnormal. Since the scene contains intense background noise (e.g., high-speed trains and heavy trucks), the proposed identification method’s complexity escalates. Nevertheless, the experimental results show that our method improves the threat detection rate by 7.6% and reduces the false alarm rate by 0.7% compared to the state-of-the-art supervised network.

## 2. Method

### 2.1. Principle of the DAS System

DAS can be understood as multiple sensors along an optical fiber (i.e., multiple continuous channels), with the distance between each channel being the spatial resolution of the DAS, and the data received by each channel being different. DAS can collect all the vibration information that causes strain on the optical fiber, such as high-speed trains, heavy vehicles passing by, knocking on isolation walls, and climbing fences. The DAS structure shown in Figure 1 is based on the preliminary work of our team [13]. This system was deployed at a high-speed railway station using optical fiber fixed to the isolation wall along the tracks. Taking the railway station as the starting point, the overall sensing range of optical fiber is about 40 km. Considering that there is a large amount of invalid data in a silent state within the actual monitoring range, only 150 channels (about 1.5 km) with abnormal events were included in the dataset construction, and the data from each expansion section forward and backward were used as experimental data. The dataset contains many different noise signals, giving the research of this area some generality.

This system applies phase-sensitive optical time domain reflectometry (Φ-OTDR) [25]. It has high-quality spatial resolution (i.e., ≥5 m) and signal-to-noise ratio, which, in turn, drastically increased the sensing range of the DAS. This system uses 1550 nm narrow linewidth fiber laser with a maximum laser power of 13 dBm. The light is divided into two parts by the 2:98 coupler (OC1), 98% of the light is modulated into pulses by an acoustic-optic modulator (AOM). EDFA is used to amplify and band pass filter (BPF) is used to suppress the amplified spontaneous emission noise. This pulse light passes through the circulator and collects Rayleigh backscattering (RB). Then the RB light is mixed with the remaining 2% of light by OC2. Subsequently, we use the balanced photodetector to detect the mixed light and retain the frequency and phase information of RB signal. The frequency and phase of the mixed signal are obtained in FPGA through fast Fourier transform. The final spatial resolution is 10.2 m and the sampling frequency (fs) of the vibration signal is 488 Hz.

### 2.2. Proposed Framework

The framework of the proposed unsupervised method is illustrated in Figure 2, which shows the vibration signal being captured and the training and testing phases. First, the DAS collects vibration signals along the optical fiber and generates space-time windows as the smallest input units.

During training, a convolutional autoencoder obtains normal features, and each input, $x_{i}$, obtains *r n*-dimensional feature vectors, $F_{i}\in R^{r\times n}$. Next, a clustering algorithm is used to establish *K* feature centers, $C\in R^{K\times n}$. During testing, the autoencoder obtains latent features from the test data, calculating the distances between the latent features and their center as the model’s output. If the distance is greater than a given threshold (using validation data), this window is identified as containing an anomaly. Finally, the results are smoothed to improve performance.

#### 2.2.1. Space-Time Window

The efficacy of the proposed method was evaluated by considering a real-world intrusion scene at a high-speed railway station. The collected dataset includes intrusion behaviors, such as climbing the isolation wall, destroying the wall, and destroying the isolation iron spikes. All of them are regarded as abnormal events during the test. Additionally, the signal strength of noise, such as that of a high-speed train, may be several times of that of abnormal events, which increases the difficulty of event recognition (The data collected by DAS is the relative phase size between adjacent channels which can also be understood as the magnitude of vibration energy). For the data to have certain visual characteristics, they are subjected to bilinear difference processing during image visualization (all the DAS images in this paper were drawn after this operation). Figure 3 shows a typical multi-channel DAS signal, in which the abscissa is the time, the ordinate is the channel number, the red box is the signal of abnormal event (Here the abnormal event is the destroying of the separation wall), and the yellow box is the additional noise. Noise 1 is the high intensity noise when the train passes, and noise 2 is generated by the surrounding environment.

Because the data collected by the DAS system have temporal and spatial continuity that affects multiple channels simultaneously, single-channel data are insufficient. Thus, the data are divided into adjustable space-time windows of multiple adjacent channels over a given period. In the experiment, the effects of different window sizes are compared. The input is scanned over a certain time interval (i.e., the scanning period). Then, a window equal to the scanning period slides across all channels. The width of the sliding window is expressed as the channel width, and they are equal. If intrusion data are found in a window, it is determined to be an intrusion window. Each space-time window is the smallest unit input to the model. Figure 4 illustrates the window generation process.

#### 2.2.2. Autoencoder

An autoencoder is a neural network the expected output of which is the input itself. It transforms a high-dimensional input into a low-dimensional product and restores it using paired encoding and decoding processes. This procedure forces the encoder to find the most effective expression for the low-dimensional version of the high-dimensional data:(1)$$\begin{array}{cc} \; & F=E(x);\\ \; & y=D(F); \end{array},$$
where *x* is the input, *y* is the output, *E* is the encoder, and *D* is the decoder; the encoder and decoder can use any network structure. The purpose is to train *E* and *D* to minimize the differences between input and output, which forces the encoder to extract the most important features of the data to improve the decoder’s performance. Because an autoencoder is non-linear, it differs from linear dimensionality reduction methods, such as principle component analysis, making it applicable to this study. The influences of different autoencoding depths on model recognition and the number of calculations are discussed in the experimental section. The loss function uses mean square error:(2)$$loss\left(x,y\right)=\frac{1}{N}\sum {∥x-y∥}^{2},$$
where *N* is the number of windows, and Adam optimization is used to update the network parameters. The specific structure is given in the experiment section.

#### 2.2.3. Clustering

A clustering algorithm is a typical unsupervised learning method in which similar data are apportioned in groups without regard to class labels. The K-means [26] version is widely used, owing to its simplicity and efficiency. A smaller Euclidean distance between two targets means greater similarity.

In this study, the autoencoder was used to extract normal features, and K-means clustering was used to establish 128 cluster centers.

#### 2.2.4. Model Recognition

During testing, the distances between features and their resultant centers were computed using Equation (Equation 3). $F_{i}^{j}\in R^{n},j\in \left\{1,\ldots ,r\right\}$ is a latent input feature ($x_{i}$):(3)$$\begin{array}{cc} \; & k^{\prime }=\underset{k}{argmin}{∥F_{i}^{j}-C^{k}∥}^{2};\\ \; & d^{j}={∥F_{i}^{j}-C^{k^{\prime }}∥}^{2}. \end{array}$$

The largest *m* distances were averaged as the output value of each window:(4)$$D_{i}=\frac{1}{m}\sum_{l=1}^{m}Top_{l}\left(d\right),$$
where $Top_{l}$ is a function that determines the *l*-th largest distance. If the output value is greater than the threshold, the corresponding window is considered an anomaly:(5)$$s_{i}=\left\{\begin{array}{cc} 1, & D_{i}>TH\\ 0, & others \end{array}\right.$$
where TH is the threshold set by the ratio of the number of false positive windows in the validation set, and $s_{i}$ is set to one if $x_{i}$ is anomalous; otherwise, it is zero.

#### 2.2.5. Window Accumulation

Because anomalous behaviors are continuous, those of short duration are likely to produce false positive detection. To avoid this, a window accumulation mechanism is added after the output. When determining whether a window, $s_{i}$, is anomalous, the current and prior α windows are traversed in retrograde. If the anomalous window exceeds β, $s_{i}$ is regarded as a true positive. Given multiple windows containing the same set of channels, $s_{1},s_{2},\ldots$, the final test result, $s_{i}^{\prime }$, is defined as
(6)$$s_{i}^{\prime }=\left\{\begin{array}{cc} 1, & \sum_{j=i-\alpha +1}^{i}s_{j}\geq \beta \\ 0, & others \end{array}\right.,0<\beta \leq \alpha \leq i.$$

#### 2.2.6. Threshold Setting

The threshold is set on the validation set to detect space-time window abnormalities. If the distance is greater than this threshold, the window is marked as abnormal. The threshold set on the verification set is the calibration of the actual environment. Subsequent experiments are conducted on the test set using this threshold.

## 3. Experiment

### 3.1. Evaluation Metrics and Parameters

The threat detection rate (TDR) is the ratio of true positive intrusion windows detected after the window accumulation is equivalent to the recall rate:(7)$$TDR=\frac{TP}{TP+FN}=recall,$$
where TP is a true positive, and FN is a false negative.

The false alarm rate (FAR) is the ratio of false alarms after the window accumulation is equivalent to the one-minus-precision rate:(8)$$FAR=1-\frac{TP}{TP+FP}=1-precision.$$

FP is a false positive. The $F1_{score}$ is used to evaluate the balance between intrusion detection accuracy and recall rate; the higher the $F1_{score}$, the better:(9)$$F1_{score}=\frac{2*precision*recall}{precision+recall}=\frac{2*(1-FAR)*TDR}{(1-FAR)+TDR}.$$

The $F1_{score}$ objectively measures overall performance according to the false positive and intrusion detection rates. A higher $F1_{score}$ is obtained only when the intrusion detection rate is high, and the false positive rate is low. The response time (RT) is the average time between the space-time window at the beginning of the true-positive intrusion signal at the predicted value.

Several settable parameters can be used to fine-tune model performance. The threshold (TH) determines whether the space-time window is indeed an intrusion window. If the distance of the event from the center is greater than this threshold, it is determined to be an intrusion window. The threshold setting is used to control model sensitivity. Unless otherwise specified, the threshold is obtained when the false alarm rate of the validation set is 3%. The scanning period controls the temporality of the intercepted space-time window, whereas the channel width controls its spatiality. Variable *m* is the average of the largest distances; thus, model sensitivity can be controlled by adjusting *m*. Unless otherwise specified, *m* is set to 64. α is the number of windows traversed in a retrograde strategy to control model sensitivity. Last, β is the window threshold that triggers judgment in the window accumulation strategy, which also controls model sensitivity.

### 3.2. Experimental Set Up

First, the DAS system illustrated in Figure 1 was used to collect vibration signals, in which the DAS spatial resolution was set to 10 m (i.e., the space interval between two channels), and the signal sampling rate was 488 Hz. The DAS system was deployed at a high-speed railway station, and the optical fiber was fixed along the barrier beside the tracks for approximately 40 km.

In a normal environment, 60 min signal data were randomly and intermittently collected and 40 min were randomly selected for the training set. The other data were used for validation. The test data were the same as those used in [13], which included approximately 30 min of multiple intrusion behaviors. The number of windows generated is shown in Table 1 (The ratio of abnormal to normal data is about 1:10). To eliminate the bias between the various channels and the effect of some too large data on the results, before being input to the model, all channels were standardized at each moment, and the maximum value of each channel was set to 3.5 times the standard deviation of the same channel in the training set. Finally, the vibration data were normalized in the [0, 1] range.

### 3.3. Influence of Model Parameters

#### 3.3.1. Influence of Model Depth

Different model depths are listed in Table 2, and Bold in the table represents the best result for this item. The autoencoder model was used to extract latent features. The encoder structure of each layer was a Conv-ReLU-BatchNorm-Maxpool, and the decoder structure of each layer was an Interpolate-Conv-ReLU/Sigmoid, the convolution kernel is 3. Each encoder layer corresponded to one decoder layer. Considering the model result and calculation time, autoencoder models with 1–3 layers were compared. The results showed that the autoencoder using only one convolutional layer achieved the best performance, and owing to the shallow structure, the model saved computation time.

#### 3.3.2. Influence of Scanning Period and Channel Width

The scanning period was used to control the space-time window and the channel width was used to control the spatial range. When experimentally compared, the window-scanning period range was 128∼4096, and the channel widths range was 10∼35.

The table shows that model detection accuracy improved with the increasing scanning period. In particular, the FAR dropped significantly. The response time increased accordingly, as a longer scanning period caused an increased delay in model output, as detailed in Table 3.

Increasing the channel width improved anomaly detection performance and lowered the FAR. Additionally, owing to the fixed scanning period, the change in RT was small. When the channel width was too small, the missing spatial relationships led to poor detection. However, when the channel width was much larger a large amount of useless interference information was extracted and TDR dropped. A too-large channel width affected the accurate positioning of anomalies. When the channel width approximated the width of the channel affected by the abnormal behavior, the model reached its peak performance, as detailed in Table 4.

#### 3.3.3. Impact of Window Accumulation Strategy

The window accumulation was controlled by two parameters, α and β, which balanced model sensitivity and detection accuracy. The value of α was varied from 2 to 10 and that of β from 1 to α−1 to examine their influence on the model’s sensitivity and adopted a scanning period of 128.

With the increase in α, model sensitivity increased along with the rate of anomaly detection. In contrast, with the increase in β, the anomaly detection rate declined, as shown in Figure 5. The false detection of the model increased with α, and it decreased as β increased, as shown in Figure 6.

Considering the trade-off performances of TDR and FAR, the F1 scores were compared, as shown in Figure 7.

#### 3.3.4. Impact of Maximum Average Distance

Adjusting *m* can be used to tune model performance, as shown in Table 5. With an increase in *m*, various model indicators were improved. When *m* reached a reasonable value range, the TDR also improved, and the FAR decreased. Concomitantly, a larger window width requires a larger *m*.

### 3.4. Model Performance Comparison

To verify the model’s performance, experiments were conducted on this dataset using a variety of supervised algorithms as the baseline. The results are shown in Table 6. The proposed method was trained using normal data and the supervised methods were trained on labeled anomaly data of the same size, as were the other configurations.

Table 6 shows that the proposed method’s anomaly detection rate was approximately 1% lower than those of ConvLSTM and DenseNet but above those of other models. The FAR was only 0.7% higher than the lowest model, Shiloh18, while the F1 score was higher than most supervised models and only 0.6% lower than ConvLSTM. The poor performance of traditional machine learning methods (e.g., LR and LightGBM) showed that deep learning methods have certain advantages in processing DAS signal data. Overall, the proposed model outperformed most supervised networks and was second only to ConvLSTM. Table 7 presents the post-tuned comparisons with ConvLSTM using the same channel widths. In addition, directly applying unsupervised methods(e.g., Chen and Ji) from other fields to DAS can lead to a high false alarm rate

With a similar RT, the TDR of the proposed model was 7.6% higher than that of ConvLSTM. The FAR was 0.7% lower, and the F1 score was 4.5% higher than ConvLSTM. The experimental results show that unsupervised anomaly detection methods can achieve better performances than state-of-the-art methods.

### 3.5. Visualization of Results

As one of the test samples, the visualization of model detection shown in Figure 3 is shown in Figure 8. Nearly all intrusion signals were detected even with strong noise interference.

The results after window smoothing are shown in Figure 9, and further confirm the model detection performance. Window smoothing also eliminated the false positives caused by single-window identification errors.

As shown in Figure 10, the scanning period produced 1024 results. Furthermore, the false positives reduced significantly as the scanning period increased.

## 4. Conclusions

This study developed a label-free anomaly detection method based on DAS, which, to detect a variety of anomalous events, required only normal-state data for training. Because this method does not require labeled abnormal data, it addresses the problem that labeled anomaly data are generally lacking and sidesteps the impossibility of defining all types of anomalies for supervised networks. In particular, the proposed lightweight model reduces several parameters and vast amounts of computation.

The proposed method was validated using a high-speed rail intrusion dataset and was compared with multiple supervised methods under the same experimental configuration. The results show that the threat detection rate of this method reaches 91.5%, which is 5.9% higher than that of the state-of-the-art supervised network, and the false alarm rate reaches 7.2% is 0.8% lower than the supervised network.

We intend to test the model in complex environments (e.g., rain and thunder). By evaluating the efficacy of the proposed approach in more complex environments, the model will be refined for even better performance.

## Figures and Tables

**Figure 1 sensors-23-04094-f001:**
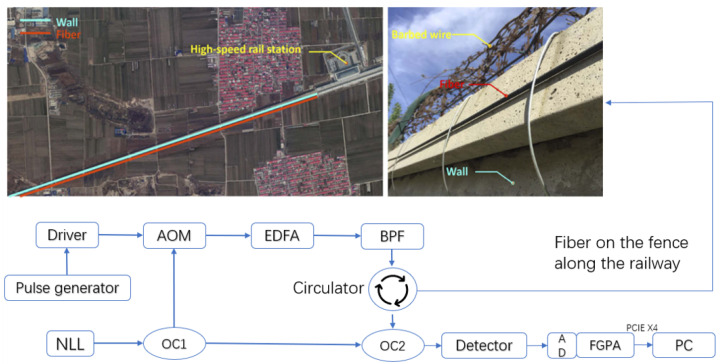
Optical setup of the DAS system.

**Figure 2 sensors-23-04094-f002:**
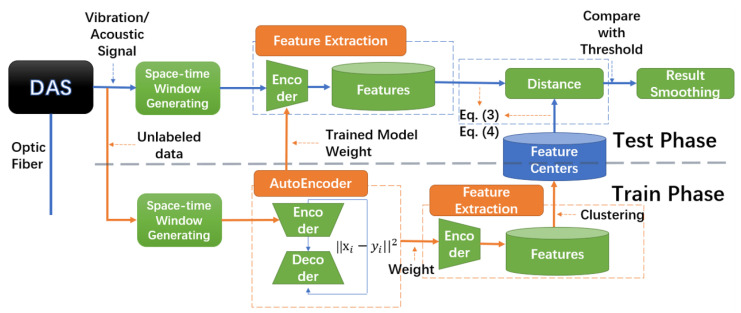
The framework of the label-free anomaly detection method.

**Figure 3 sensors-23-04094-f003:**
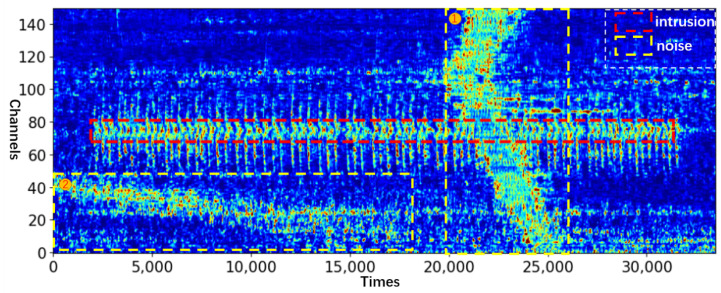
DAS signals of destroying the separation wall (fs = 488 HZ).

**Figure 4 sensors-23-04094-f004:**
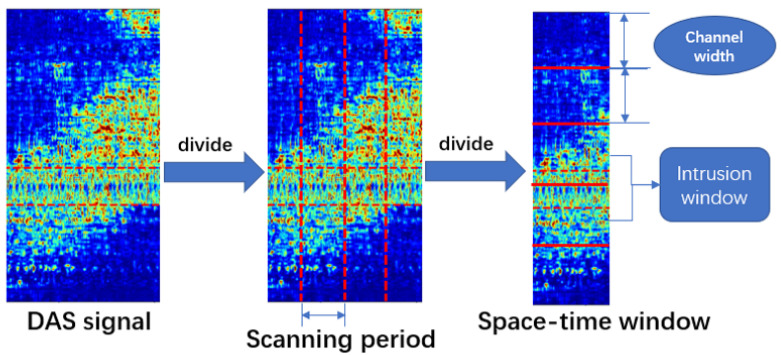
Generate space-time windows.

**Figure 5 sensors-23-04094-f005:**
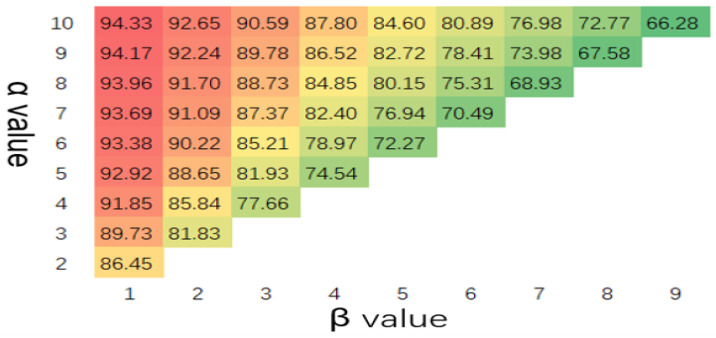
Threat detection rate(%) with different values of α and β (scanning period = 128 and channel width = 20).

**Figure 6 sensors-23-04094-f006:**
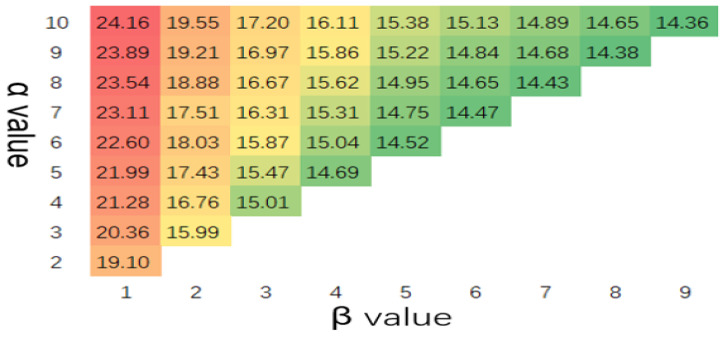
False alarm rate(%) with different values of α and β (scanning period = 128 and channel width = 20).

**Figure 7 sensors-23-04094-f007:**
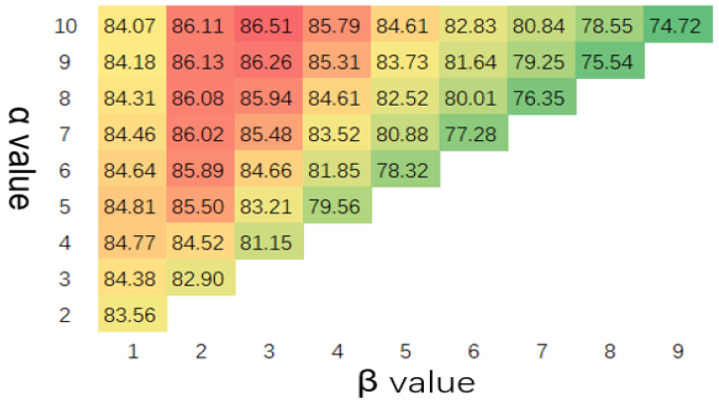
F1 score(%) with different values of α and β (scanning period = 128 and channel width = 20).

**Figure 8 sensors-23-04094-f008:**
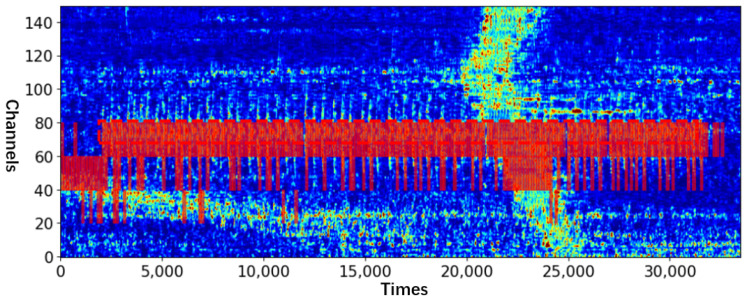
Before the accumulation of the model test result window, the red boxes are the detected intrusion windows (scanning period = 128 and channel width = 20).

**Figure 9 sensors-23-04094-f009:**
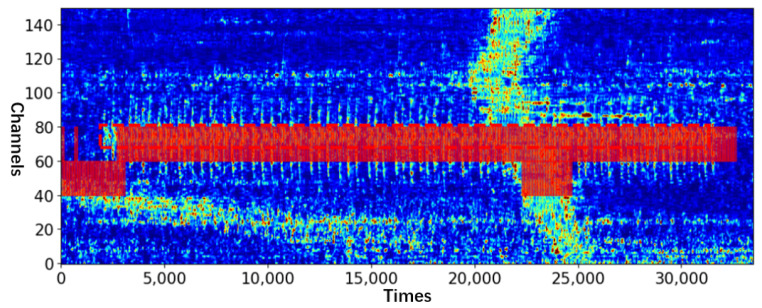
After the accumulation of the model test result window, the red boxes are the detected intrusion windows.

**Figure 10 sensors-23-04094-f010:**
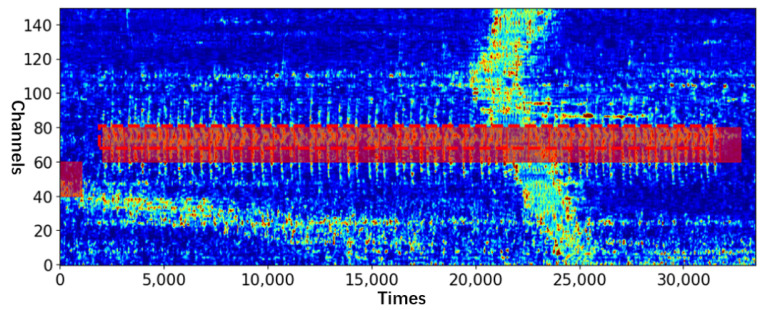
After the accumulation of the model test result window, the red boxes are the detected intrusion windows (scanning period = 1024 and channel width = 20).

**Table 1 sensors-23-04094-t001:** Number of windows generated with different scanning period and channel width (The ratio of abnormal to normal data is about 1:10).

Scanning Period	128	256	512	1024	2048	4096
W = 10	296,603	147,668	73,502	36,221	17,634	8415
W = 15	186,293	92,719	46,126	22,711	11,040	5262
W = 20	135,120	67,243	33,444	16,459	7996	3800
W = 25	108,900	54,173	26,929	13,237	6414	3043
W = 30	88,546	44,048	21,892	10,761	5221	2479
W = 35	65,038	32,339	16,055	7877	3815	1804

**Table 2 sensors-23-04094-t002:** Influence of autoencoder model depth, “Par” represents the model parameter quantity, “FLOPs” represents floating point operations.

Depth of Model	TDR	FAR	F1	Par(K)	FLOPs(B)
1	**90.6%**	**17.2%**	**86.5%**	**1.34**	**3.44**
2	90.1%	22.5%	83.3%	47.20	31.96
3	83.2%	24.6%	79.1%	314.72	60.11

**Table 3 sensors-23-04094-t003:** The anomaly detection results under different scanning period (the channel width is 20).

Scanning Period	TDR	FAR	F1	RT
128	90.6%	17.2%	86.5%	**0.86s**
256	90.3%	16.8%	86.5%	1.99s
512	79.4%	13.5%	82.8%	4.93s
1024	83.9%	12.5%	85.6%	4.96s
2048	80.8%	9.1%	85.5%	7.38s
4096	**91.5%**	**7.2%**	**92.1%**	8.51s

**Table 4 sensors-23-04094-t004:** The anomaly detection results under different channel widths (the scanning period is 4096).

Channel Width	TDR	FAR	F1	RT
10	69.1%	16.8%	75.5%	9.57s
15	76.7%	11.9%	81.6%	10.16s
20	**91.5%**	7.2%	**92.1%**	**8.51s**
25	67.9%	13.5%	76.1%	9.69s
30	79.1%	**6.3%**	85.7%	10.40s
35	51.3%	19.4%	62.7%	13.47s

**Table 5 sensors-23-04094-t005:** The anomaly detection results under different m (scanning period = 4096 and channel width = 20).

m	TDR	FAR	F1	RT
1	84.5%	10.0%	87.1%	9.69s
16	89.6%	8.1%	90.7%	9.22s
64	91.5%	**7.2%**	92.1%	**8.51s**
256	93.2%	7.3%	**93.2%**	8.63s
1024	**94.1%**	7.7%	93.1%	8.51s
4096	91.3%	11.8%	89.6%	8.62s

**Table 6 sensors-23-04094-t006:** Comparison of anomaly detection performance.

Algorithm Model	TDR	FAR	F1
Shiloh18 [27]	69.6%	**11.8%**	77.8%
Aktas17 [28]	78.1%	13.0%	82.3%
DenseNet [29]	85.0%	21.2%	81.8%
LSTM [30]	77.4%	34.3%	71.0%
A-LSTM [31]	63.9%	27.2%	68.1%
LR [32]	37.2%	41.4%	45.5%
LightGBM [33]	41.2%	35.0%	50.5%
ConvLSTM [13]	**85.1%**	12.6%	**86.2%**
Chen [34]	68.6%	24.0%	72.1%
Ji [35]	82.3%	21.6%	80.3%
**Our Method**	83.9%	12.5%	85.6%

**Table 7 sensors-23-04094-t007:** The anomaly detection results of our model and ConvLSTM (the channel width = 20).

Algorithm Model	TDR	FAR	F1	RT	Par(K)
ConvLSTM [13]	85.6%	8.0%	88.7%	**8.25s**	79.55
**Our Method**	**91.5%**	**7.2%**	**92.1%**	8.63s	1.34

## Data Availability

Due to privacy restrictions, the dataset has not been published yet. You can contact the corresponding author to obtain.

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
