# Peer review of "Label-Free Anomaly Detection Using Distributed Optical Fiber Acoustic Sensing"

_sensors, 2023, doi:10.3390/s23084094_

Round 1

Reviewer 1 Report

The reported results on unsupervised anomaly detection in DAS signal data are interesting and worthy of publication by Sensors.

There are some English text to be corrected:

p. 5/lines 135-136: instead of '...that apportions ...' better is: in which similar data are apportioned in groups ...

p.9/line251: Better write: but above those of other models.

p. 10/line 270: Fig10the ---> Fig. 10, the

p. 10: line 272 should be deleted.

p. 11/line 275 instead of requires: required

Caption of Fig. 4: Generating space-time windows. 

Captions of Figs. 8-10: boxs ---> boxes

Author Response

Thank you for pointing out errors in the English text, and we apologize for these low-level errors. We have read the entire text and corrected all listed errors. In addition, we have made the following changes:

1. We have revised the dimensions of the figures. 5-7 to make them clearer.

2. We modified the text size of Figures 3, 4, 8, 9, and 10 to match the text size of the article.

Reviewer 2 Report

The author has proposed an unsupervised deep-learning method that only learns the normal data features from ordinary events. A convolutional autoencoder is used to extract DAS signal features. A clustering algorithm then locates the feature center of the normal data, and the distance to the new signal is used to determine whether it is an anomaly.

The DAS was used to collect vibration signals, in which the DAS spatial resolution was set to 10 m (i.e., the space interval between two 189 channels), and the signal sampling rate was 488 Hz. The DAS system was deployed at a high-speed railway station, and the optical fiber was fixed along the barrier beside the 191 tracks for approximately 40 km.

The paper is interesting work for the reader in this area. The author has to address my comments as follows;

1.       What type of vibration signal can be recorded by DAS?

2.       How much range of the vibration frequency can be captured?

3.       How about the resolution of the DAS performance?

4.       In conclusion, it should be mentioned about the results.

Author Response

1.What type of vibration signal can be recorded by DAS?

Response:Thanks for your reminding. We have provided a more detailed description of the types of vibration signals that DAS devices can collect in the line 69-71 on page 2:

DAS can collect all the vibration information that causes strain on the optical fiber such as high-speed trains, heavy vehicles passing by, knocking on isolation walls, and climbing fences.

And the deployment of DAS in the line 72 on page 2:

This system was deployed at a high-speed railway station using optical fiber fixed to the isolation wall along the tracks.

Therefore, DAS can collect all vibration information near the isolation wall.

2.How much range of the vibration frequency can be captured?

3.How about the resolution of the DAS performance?

Response:

Thank you for your comment. We believe that these two issues can be summarized as the specific settings of the DAS in our experiment. A description of DAS is helpful for the readers to understand our work. The optical configuration of the DAS system was described in the line 79-92 on page 3, and the resolution of the DAS performance in the line 91-92:

The spatial resolution of the DAS device set in our experiment is 10.2 m, and the sampling frequency of the vibration signal is 488 Hz.

4.In conclusion, it should be mentioned about the results.

Response: Thank you for reminding us. We did not provide a clear summary of the work in this article in the conclusion, so we added a description in the line 285-288 on page 10:

The proposed method was validated using a high-speed rail intrusion dataset and was compared with multiple supervised methods under the same experimental configuration.The results show that the threat detection rate of this method reaches 91.5\%, which is 5.9\% higher than that of the state-of-the-art supervised network, and the false alarm rate reaches 7.2\% is 0.8\% lower than the supervised network.

Reviewer 3 Report

The authors proposed a solution of label-free anomaly detection using distributed optical fiber acoustic sensing based on unsupervised learning, which does not require labeled data for training. However, there are still some concerns that required extra explanation.

1. The definition of anomaly is not clear, it is supposed to give some examples;

2. The comparison in the experiment is not sufficient to prove the proposal;

3. The unit used in the experiment is not given.

Author Response

1.The definition of anomaly is not clear, it is supposed to give some examples;

Response: Thank you for reminding us. We may not have explained clearly what the anomaly to be detected in the text is. We have added a description to the summary in the line 9-10:

The efficacy of the proposed method was evaluated in a real high-speed rail intrusion scenario, and consider 9 all behaviors that may threaten the normal operation of high-speed trains as abnormal.

And We carefully described Figs. 3(DAS signals of destroying the separation wall).  Details as follows:

1)Caption of Fig. 3 on page 4 is revised as:

DAS signals of destroying the separation wall(fs=488HZ)

2)Added text description:

Figure 3 shows a typical multi-channel DAS signal, in which the abscissa is the time, the ordinate is the channel number, the red box is the signal of abnormal event (destroying the separation wall), and the yellow box is the additional noise. Noise 1 is the high intensity noise when the train passes, and noise 2 is generated by the surrounding environment.

3.The comparison in the experiment is not sufficient to prove the proposal;

Response: Thanks for your comments. In response to this opinion, we have added two unsupervised methods to the comparative experiment in Table 6 to prove the progressiveness of our method on page 10:

In addition, directly applying unsupervised methods(e.g., Chen and Ji) from other fields to DAS can lead to a high false alarm rate

4.The unit used in the experiment is not given.

Response:Thank you for your reminder. The data collected by DAS is the relative phase size between adjacent channels. The basic unit of this data is ‘rad’, but after some comparative calculations, there is no clear unit. The magnitude of this value represents the strain level of the optical fiber, which can also be understood as the magnitude of vibration energy. Added description in the line111-113 on page 4.

Reviewer 4 Report

In this article, the authors developed a label-free anomaly detection method based on DAS. In their scheme, an unsupervised deep-learning method that only learns the normal data features from ordinary events is proposed and validated using a high-speed rail intrusion dataset. This method does not require labeled abnormal data, it addresses the problem that labeled anomaly data is generally lacking and sidesteps the impossibility of defining all types of anomalies for supervised networks. In addition, the proposed lightweight model reduces several parameters and amounts of computation. This is a study with good practical application value, but there are some suggestions.

  1. It is recommended that Fig. 5- Fig. 7 to be displayed in a tabular format. Fig.3, Fig.4, Fig. 8, Fig.9 and Fig. 10 are not well formatted, the size of the axis headings and text is too large.

2. What is the specific relationship between the actual sensing distance and the channel?

3. The author claims that the overall sensing range was approximately 40km, so it is suggested to give the time-domain diagram of DAS output at the sensing distance of 40km.

4. I have noticed that the system response time is about 8 seconds, which means that the actual vibration frequency can be measured is about 0.125Hz. Is this valuable for practical applications?

Author Response

1.It is recommended that Fig. 5- Fig. 7 to be displayed in a tabular format. Fig.3, Fig.4, Fig. 8, Fig.9 and Fig. 10 are not well formatted, the size of the axis headings and text is too large.

Response: Thank you for pointing out these errors. We are very sorry for making these low-level mistakes. But we believe that using images can better indicate the trend of data changes of Fig. 5- Fig. 7. However we have modified the size of the image to make it clearer. And the text dimensions of Fig. 3, 4, 8, 9, and 10 have been adjusted.

2.What is the specific relationship between the actual sensing distance and the channel?

Response: Thanks for your comments. We didn't clearly articulate the relationship between the two, therefore, we have added a narrative to the text in the line 67-69:

DAS can be understood as multiple sensors along an optical fiber (i.e. multiple continuous channels), with the distance between each channel being the spatial resolution of the DAS, and the data received by each channel being different.

And the spatial resolution of our DAS is 10.2 m in the line 90.

3.The author claims that the overall sensing range was approximately 40km, so it is suggested to give the time-domain diagram of DAS output at the sensing distance of 40km.

Response: Thank you for reminding us, perhaps the expression in the original text is not clear enough, the overall sensing range of our DAS is about 40 km. However, it contains a large amount of meaningless data, and our dataset only contains approximately 1.5 km, including abnormal events and some forward and backward expanding data.  

Modify in the line70-75,as follows:

Taking the railway station as the starting point, the overall sensing range of optical fiber is about 40 km. Considering that there is a large amount of invalid data in a silent state within the actual monitoring range, only 150 channels (about 1.5 km) including abnormal events were included in the dataset construction, and the data from each expansion section forward and backward were used as experimental data.

4.I have noticed that the system response time is about 8 seconds, which means that the actual vibration frequency can be measured is about 0.125Hz. Is this valuable for practical applications?

Response: Thanks for your comments. The calculation in the article is the worst-case response time, and in fact, abnormal events can be correctly identified even if they occur in the second half of a scanning cycle which will greatly shorten the response time. We have added an experiment to test the detection rate of abnormal events in shorter response times using sliding windows.

The response time is from the occurrence time of abnormal events to the end of the sapce-time window. And the experimental results show that even if RT is reduced to 1 second, the detection accuracy can reach 88%

Round 2

Reviewer 3 Report

The authors have modified the manuscript as suggested.